# Parameterized and Consistency Tests of Gravity with Gravitational Waves: Current and Future †

## Zack Carson and Kent Yagi *

Department of Physics, University of Virginia, Charlottesville, VA 22904, USA
* Correspondence: ky5t@virginia.edu
† Presented at the Recent Progress in Relativistic Astrophysics, Shanghai, China, 6–8 May 2019.

**Abstract:** Gravitational wave observations offer unique opportunities to probe gravity in the strong and dynamical regime, which was difficult to access previously. We here review two theory-agnostic ways to carry out tests of general relativity with gravitational waves, namely (i) parameterized waveform tests and (ii) consistency tests between the inspiral and merger-ringdown portions. For each method, we explain the formalism, followed by results from existing events, and finally we discuss future prospects with upgraded detectors, including the possibility of using multi-band gravitational-wave observations with ground-based and space-borne interferometers. We show that such future observations have the potential to improve upon current bounds on theories beyond general relativity by many orders of magnitude. We conclude by listing several open questions that remain to be addressed.

**Keywords:** gravitational waves; black holes; general relativity; modified theories of gravity

## 1. Introduction

Einsteins' famous theory of general relativity (GR) has proven to be wildly successful for over 100 years, accurately predicting many astrophysical phenomena observed to this very day. Throughout this period of time, many have attempted to prove the theory incorrect or merely just one piece of a more grand theory of nature with various observational and experimental schemes. All have met with the same result: GR still standing true with absolutely no statistically significant signs of deviation. With such an outstanding history of success, why must we continue to test the theory of GR? The answer is simple: There yet remains a plethora of unanswered questions stemming from mysterious observations seen throughout this time. These open questions include, and are not limited to, the unification of GR and quantum mechanics [1–6], dark matter and its influence on galactic rotation curves [1–4,7], dark energy and the ensuing accelerated expansion of the universe [2,5,6,8], the strange inflationary period seen in the early universe [1–3,6], and the matter-anti-matter asymmetry found in the present universe [1,3]. To date, there have been several proposed theories of gravity, both modifications or extensions to GR, as well as entirely new theories, many of which could be used to explain several of the above-listed open astrophysical/cosmological questions. While these theories could potentially reduce to the GR we know in the weak-field environments typically observed, they could very well become active in the extreme-gravity regime where the gravitational fields are strong, non-linear, and highly-dynamical.

For the last century, many attempts have been made to determine and constrain the various proposed modified theories of gravity found in the literature. When probed in the weak-field and static environments such as the local solar system, observations of photon-deflection, Shapiro time-delay, perihelion advance of Mercury, the Nordtvedt effect, and more [9] have determined no deviations from GR. Similarly, observations concerning the strong-field and static systems

of binary pulsar systems [10,11] have also shown to be consistent with GR. Further, large-scale cosmological observations [1,2,6,8,12] have also identified no deviations. More recently, the groundbreaking gravitational wave (GW) observations of coalescing black holes (BHs) [13,14] and neutron stars (NSs) [15] by the Laser Interferometer Gravitational-Wave Observatory (LIGO) and Virgo Collaborations (LVC) have provided us with the unique opportunity to study fascinating extreme-gravity environments. To date, all such confirmed events have similarly found no deviations from GR [16,17].

While the current extreme-gravity tests of GR have yet to discover ground-breaking results, hope is not lost, as the field of gravitational wave astronomy is still in its infancy. While monumental in their engineering design and successful sensitivity, the current LVC Observing Run 2 (O2) infrastructure is limited by noise. Due to the LVC's overwhelming successes on the GW front, several proposed, planned, and even funded gravitational wave detectors are in the works. Several planned upgrades to the current LIGO detectors, aLIGO, A+, and Voyager [18,19], are currently underway with large improvements in the design sensitivity. Furthermore, new "third generation" interferometers Cosmic Explorer [19] (CE), and the Einstein Telescope [20] (ET) with up to $\sim 100$ times the sensitivity of current detectors are currently in the planning stages. Finally, space-based laboratories, such as the Laser Interferometer Space Antenna (LISA) [21], TianQin [22,23], the Deci-Hertz Interferometer Gravitational wave Observatory (DECIGO) [24], and B-DECIGO [25], are currently in progress with sensitivities to GWs in the sub-Hz frequency bands. For stellar-mass binary BHs, ground-based detectors sensitive to high GW frequencies can largely probe non-GR effects, as they become more active at high relative velocities; while space-based detectors that are operative at low frequencies, are more suited to probing low-velocity effects. With such a promising future of observational GW astrophysics, probes of modified theories of gravity stand a highly increased likelihood of observing possible deviations from GR.

In the following document, we summarize the past, present, and future considerations for testing GR in the extreme gravity environments of merging BHs. In particular, we consider both parameterized tests of GR and consistency tests between the inspiral and merger-ringdown portions of the GW signal. The former tests allow one to map generalized non-GR effects entering the gravitational waveform with a certain velocity dependence, to most proposed modified theories of gravity and their associate theoretical parameters. The latter allows one to test how consistent the obtained signal is with the predictions of GR as a whole, granting a gauge on how much the inspiral and merger-ringdown signals agree. Specifically, we present current and projected bounds on the Einstein dilaton Gauss–Bonnet (EdGB) [26,27], dynamical Chern–Simons (dCS) [28–31], scalar-tensor theories [32,33], noncommutative theories [34,35], time-varying $G$ theories [9,36,37], time-varying BH mass theories [38,39], and massive gravity [40–43]. We discuss the current constraints and progress, followed byy estimated future bounds. The latter is considered from single-band detections on both ground- and space-based GW detectors, as well as the multi-band observations between both detector types. We find that orders-of-magnitude improvements can be made upon using such future considerations, for both parameterized and consistency tests of GR.

The organization of this article is as follows. Section 2 details the parameterized tests of GR, starting off with the formulation and techniques used, followed up with the current status and future predictions of constraints on non-GR effects, finishing up with a discussion of multi-band observations. Section 3 follows suit with the same organization for the inspiral-merger-ringdown consistency tests of GR. Finally, in Section 4, we conclude and discuss the open questions yet remaining in the testing of non-GR effects. Throughout this document, we utilize the geometric units of $G = 1 = c$, unless otherwise stated.

## 2. Parameterized Tests

### 2.1. Formulation

Instead of comparing GW data with template waveforms in specific modified theories of gravity one by one, a more efficient approach is to first compare the data with template waveforms that can capture generic non-GR modifications, and then map the information from generic non-GR parameters to that of parameters in each theory. Various formalisms exist for such a theory-agnostic approach [16,17,44–49]. Here we follow the parameterized post-Einsteinian (ppE) formalism [47], in which the frequency-domain waveform is given by

$$\tilde{h}(f) = \tilde{h}_{\mathrm{GR}}(1 + \alpha \, u^a)e^{i\delta\Psi}, \quad \delta\Psi = \beta u^b. \tag{1}$$

Here $\tilde{h}_{\mathrm{GR}}$ is the GR waveform (for which we use the IMRPhenomD waveform [50,51] for spin-aligned binary black holes (BBHs) with circular orbits) while $u = (\pi\mathcal{M}f)^{1/3}$ is the effective relative velocity of binary constituents with $\mathcal{M}$ and $f$ representing the chirp mass and GW frequency respectively. $(\alpha, a, \beta, b)$ are known as the ppE parameters. $\alpha$ and $\beta$ denote the overall magnitude of the non-GR term in the amplitude and phase respectively, while $a$ and $b$ characterize at which post-Newtonian (PN)[1] order the correction enters the gravitational waveform in the amplitude and phase. The mapping between these ppE parameters and theoretical constants in various modified theories of gravity can be found in Tables I and II of [37].

Let us now prepare the basics of the Fisher analysis methods used frequently in this document for parameter estimation of template parameters $\theta^a$. Commonly used as a less-computationally expensive alternative to a full Bayesian statistical analysis, the Fisher analysis is a good approximation for loud enough events. The signal-to-noise-ratio (SNR) $\rho$ of such events is defined as

$$\rho \equiv \sqrt{(h|h)}, \tag{2}$$

where $h$ is the gravitational waveform template, and the inner product $(a|b)$ is defined to be

$$(a|b) \equiv 2 \int_{f_{\mathrm{low}}}^{f_{\mathrm{high}}} \frac{\tilde{a}^*\tilde{b} + \tilde{b}^*\tilde{a}}{S_n(f)} df. \tag{3}$$

In the above expression, $S_n(f)$ represents the spectral noise density of the given detector, and $f_{\mathrm{high,low}}$ are the cutoff frequencies, again dependent on the detector.

Assuming a Gaussian-distributed noise pattern, and Gaussian prior distributions on waveform template parameters, the parameters $\theta^a$ assuming a GW signal $s$ can be found to follow [52]

$$p(\theta^a|s) \propto p_{\theta^a}^{(0)} \exp\left[-\frac{1}{2}\Gamma_{ij}\Delta\theta^i\Delta\theta^j\right]. \tag{4}$$

In the above distribution, $\Delta\theta^i \equiv \theta^i - \hat{\theta}^i$ with $\hat{\theta}^i$ representing the maximum likelihood value of $\theta^i$, $p_{\theta^a}^{(0)}$ is the prior probability distribution which we assumed to be Gaussian with root-mean-square errors $\sigma_{\theta^a}^0$, and $\Gamma_{ij}$ is the Fisher information matrix determined to be

$$\Gamma_{ij} \equiv (\partial_i h|\partial_j h). \tag{5}$$

---

1　　A term of $n$PN order in the waveform is proportional to $(u/c)^{2n}$ relative to the leading-order term.

The resulting $1\sigma$ root-mean-square errors on template parameters $\theta^a$ can be written directly as

$$\Delta\theta_i \approx \sqrt{(\tilde{\Gamma}_{ii})^{-1}}, \tag{6}$$

with the effective Fisher matrix defined by [53–55]

$$\tilde{\Gamma}_{ij} \equiv \Gamma_{ij} + \frac{1}{(\sigma_{\theta^a}^0)^2}\delta_{ij}. \tag{7}$$

Finally, if one desires to combine the information from $N$ detectors, the resultant effective Fisher matrix becomes

$$\tilde{\Gamma}_{ij}^{\text{total}} = \sum_{k=1}^{N} \Gamma_{ij}^{(k)} + \frac{1}{(\sigma_{\theta^a}^0)^2}\delta_{ij}, \tag{8}$$

where $\Gamma_{ij}^{(k)}$ denotes the Fisher matrix from the $k$-th detector.

### 2.2. Current Bounds

We now review bounds on the ppE parameters from the observed GW events to date. Figure 1 presents upper bounds on $\beta$ as a function of the PN order the leading correction enters, for GW150914 and GW151226. Observe that GW151226 gives stronger bounds than GW150914 due to a larger number of GW cycles and smaller relative velocity of BHs. For comparison, we also show bounds from solar system experiments and binary pulsar observations. Notice that GW observations have an advantage on probing positive PN corrections over binary pulsar observations. The solar system bound at 1PN is much stronger than the GW bounds, though the former can only probe corrections in the conservative sector (modifications to the binding energy and Kepler's law) while the latter can probe both the conservative and dissipative sectors (GW emission).

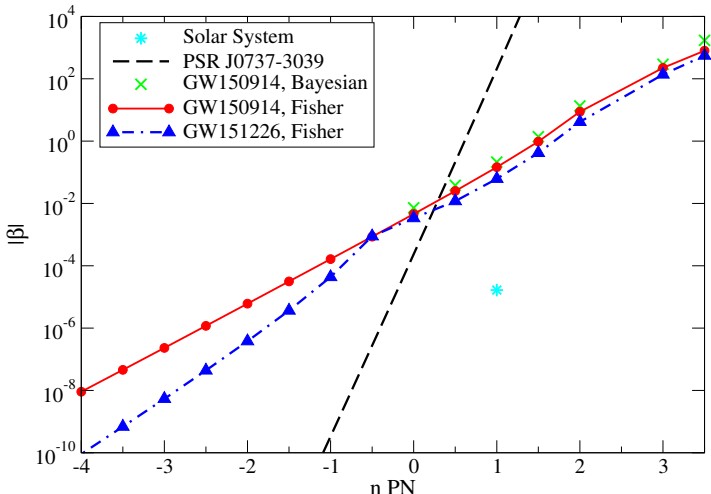

**Figure 1.** The 90% credible upper bounds on the parameterized post-Einsteinian (ppE) parameter $\beta$ at each PN order the correction enters, using solar system experiments [56] (cyan star), binary pulsar observations [57] (black dashed), GW150914 with Bayesian [17] (green crosses) and Fisher [58] (red solid) analyses, and GW151226 with a Fisher analysis [58] (blue dotted-dashed). This figure is taken from [39,58].

We next map the bounds on the ppE parameter in Figure 1 to those on example modified theories of gravity as summarized in Table 1. For example, the GW151226 bounds on EdGB are comparable to other existing bounds, while other bounds are typically weaker. However, the GW bounds have meaning as they are the first constraints obtained in the strong/dynamical field regime.

The bounds summarized in Table 1 are derived mainly from corrections in the waveform phase. We showed in [59] that amplitude corrections can give comparable bounds to those from phase corrections for massive binaries like GW150914, though inclusion of the former does not affect the bounds compared to the case where one only includes corrections in the phase, which justifies many previous works, e.g., [16,38,55,58,60–62].

**Table 1.** Each example theory (1st column) violates certain fundamental aspects of general relativity (GR) (2nd column: the strong equivalence principle (SEP), Lorentz invariance (LI), four-dimensional spacetime (4D), and massless gravitons ($m_g = 0$)) and the leading correction enters in the gravitational waveform at certain PN orders (3rd column). Each representative parameter (4th column) has been constrained from GW150914 (5th column), GW151226 (6th column) and from other observations (7th column). The top (bottom) row within massive graviton corresponds to modifications in the dynamical (propagation/conservative) sector. This table is taken and edited from [39,58].

| Theory | GR Pillar | PN | Repr. Parameters | GW150914 | GW151226 | Other Bounds |
|---|---|---|---|---|---|---|
| EdGB scalar-tensor | SEP | $-1$ | $\sqrt{|\alpha_{\text{EdGB}}|}$ [km] $\quad$ $|\dot{\phi}|$ [1/sec] | — $\quad$ — | 5.7 [31], 4.3 [59], 3.5 [63] $\quad$ $1.1 \times 10^4$ [59] | $10^7$ [64], 2 [65–67] $\quad$ $10^{-6}$ [32] |
| dCS | SEP, LI | $+2$ | $\sqrt{|\alpha_{\text{dCS}}|}$ [km] | — | — | $10^8$ [68,69] |
| Time-Varying $M$ | 4D | $-4$ | $\dot{M}$ [$M_\odot$/yr] | $\mathbf{4.2 \times 10^8}$ | $\mathbf{5.3 \times 10^6}$ | — |
| Time-Varying $G$ | SEP | $-4$ | $|\dot{G}|$ [$10^{-12}$/yr] | $\mathbf{5.4 \times 10^{18}}$ [58] $\quad$ $\mathbf{7.2 \times 10^{18}}$ [59] | $\mathbf{1.7 \times 10^{17}}$ [58] $\quad$ $\mathbf{2.2 \times 10^{16}}$ [59] | 0.1–1 [70–73] |
| Massive graviton | $m_g = 0$ | $-3$ $\quad$ $+1$ | $m_g$ [eV] | $\mathbf{6.4 \times 10^{-14}}$ $\quad$ $\mathbf{10^{-22}}$ [16,17] | $\mathbf{10^{-14}}$ [74], $\mathbf{3.1 \times 10^{-14}}$ $\quad$ $\mathbf{2.9 \times 10^{-22}}$ [17,77] | $10^{-21}$–$10^{-19}$ [75,76] $\quad$ $10^{-30}$–$10^{-23}$ [78–83] |

## 2.3. Future Bounds

Now that we have discussed the current status of parameterized tests of GR, let us now focus our attention on the future prospects of such tests [54,55,61,84–88]. Chamberlain and Yunes [86] considered the theoretical physics implications on various modified theories of gravity from BBH mergers detected by future GW detectors, which nicely complements [58] reviewed in Section 2.2. In [87,88], we similarly presented estimates on future bounds on coupling parameters for various modified theories of gravity. Here, we sum up these results for the future GW detectors Cosmic Explorer [19] (CE), LISA [21], TianQin [23], B-DECIGO [25], and DECIGO [24]. The first detector considered is a future-planned, third-generation ground-based detector with roughly $\sim 100$ times the sensitivity of the advanced LIGO design sensitivity (aLIGO) [19], and the last four are future-planned space-based detectors. The former is exceedingly efficient at probing GWs in the high frequency regime (1–$10^4$ Hz), while the latter have larger arm lengths allowing them to proficiently probe the lower frequency bands ($10^{-4}$–1 Hz for LISA and TianQin, and $10^{-2}$–$10^2$ Hz for B-DECIGO and DECIGO).

In this document, we summarize the results of [87,88], displaying constraints on the following modified theories of gravity (together with the theoretical constant and the PN order at which the leading correction enters): Einstein–dilaton Gauss–Bonnet (EdGB) gravity [26,27] ($\alpha_{\text{EdGB}}$, $-1$PN order), dynamical Chern–Simons (dCS) gravity [28–31] ($\alpha_{\text{dCS}}$, $+2$PN order), scalar-tensor theories [32,33] ($\dot{\phi}$, $-1$PN order), noncommutative gravities [34,35] ($\Lambda$, $+2$PN order), time-varying G theories of gravity [9,36,37] ($\dot{G}$, $-4$PN order), varying BH mass theories of gravity [38,39] ($\dot{M}$, $-4$PN order), massive graviton via dynamical effects [43,75] ($m_g$, $-3$PN), and massive graviton via the modified dispersion relation of the graviton [41] ($m_g$, $+1$PN order). See Berti et al. [39,60] for a comprehensive summary regarding these theories of gravity, as well as Tahura and Yagi's [37] summary of the ppE expressions used here.

Figure 2 (blue and maroon data points) displays the resulting constraints from GW150914-like events on each modified theory of gravity for future GW detectors CE, LISA, TianQin, B-DECIGO, and DECIGO. Additionally shown are the current observational constraints found in the literature. We observe that bounds on EdGB gravity can be improved upon with all four space-based detectors, dCS gravity can only be improved upon with DECIGO (further, CE, LISA, TianQin, and B-DECIGO

do not satisfy the small-coupling approximation used to derive corrections to the waveform and thus no valid bounds can be placed), noncommutative gravities can be improved upon with all five future GW detectors considered here, and massive graviton bounds (dynamical and propagation) can be improved upon only with DECIGO. In general, ground-based (space-based) detectors have the advantage on probing corrections entering at positive (negative) PN orders for GW150914-like events. See also [89] for future prospects on probing EdGB gravity and scalar-tensor theories with a mixed binary consisting of one BH and one NS.

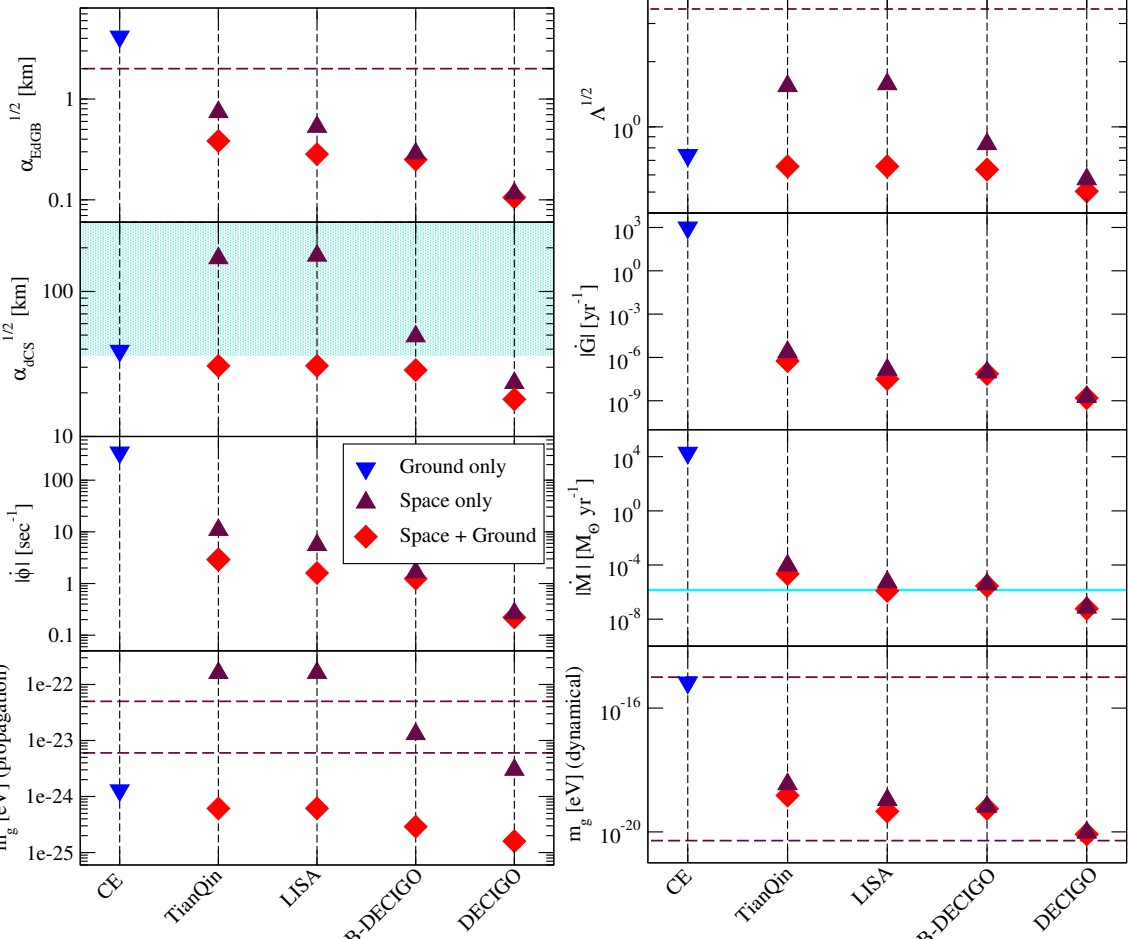

**Figure 2.** The 90% upper-bound credible level constraints on the parameters representative of the modified theories of gravity considered in [88] for GW150914-like events. Bounds are presented for Einstein dilaton Gauss–Bonnet (EdGB) gravity, dynamical Chern–Simons (dCS) gravity, scalar tensor theories, noncommutative gravity, varying-*G* theories, black hole (BH) mass-varying theories, and massive graviton (dynamical and propagation). For EdGB, dCS and scalar-tensor theories, the bounds are only meaningful outside of the blue shaded region where the small coupling approximations are violated. The dashed maroon lines correspond to the current bounds in the literature. The cyan line in the second-to-last right panel corresponds to the Eddington accretion rate: the maximum rate GW150914-like events can accrete in-falling matter under spherical symmetry. This figure is taken and edited from [88].

### 2.4. Multi-Band Bounds

In this section, we follow up the previous section by considering the combination of both observations from space and Earth, enabling the so called multi-band observations [87,88,90–93], which allows one to constrain modified theories of gravity entering at all PN orders. Following the observation of GW150914, Sesana [94] showed how joint multi-band observations of GW150914-like

events could be made with both LISA and ground-based detectors. These events would first be observed in their early inspiral stage by space-based interferometers before leaving the space-band at 1 Hz ($\sim$ 100 Hz for B-DECIGO and DECIGO) for several months before entering the ground band again to merge at $\sim$ 300 Hz, as can be seen in Figure 3. The multi-band event rates for these objects have been found to be on the order of $\mathcal{O}(1)$ by Gerosa et al. [95], due to various technical details previously unconsidered [94,96]. It was similarly shown in [96,97] that multi-band observations could be made for more massive BBHs, as well as with binary NSs [25].

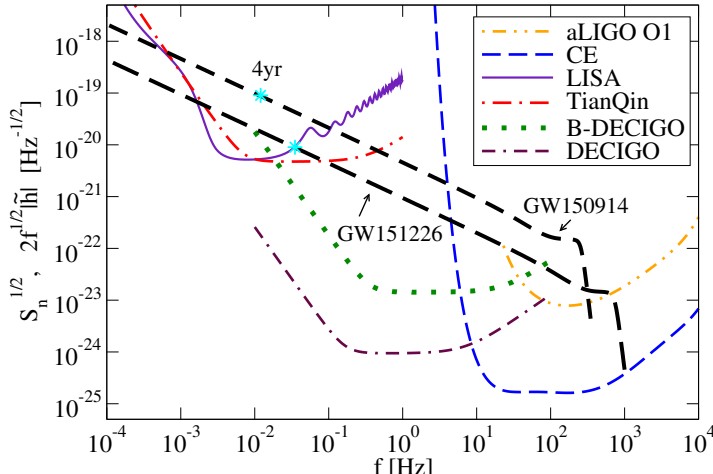

**Figure 3.** The (square root of) spectral noise densities $\sqrt{S_n(f)}$ of the gravitational-wave interferometers discussed in this document. The characteristic amplitudes $2\sqrt{f}|\tilde{h}(f)|$ for both events GW150914 and GW151226 are also displayed, with four years prior to merger shown as cyan stars. The ratio between $2\sqrt{f}|\tilde{h}(f)|$ and $\sqrt{S_n(f)}$ roughly corresponds to the signal-to-noise-ratio (SNR) of the event. Observe how the early inspiral portions of the BH coalescences are observed by the space-based detectors, while the late inspiral and merger-ringdown portions are observed by the ground-based detectors. This figure is taken and edited from [87].

In addition to providing more effective probes of gravity, multi-band observations have a myriad of other useful applications. Foremost, the early detections of binary coalescences could give alert to both ground-based detectors and electromagnetic telescopes for follow-up observations of the merger-ringdown event [94]. The former will also allow one to optimize ground-based GW detector sensitivities to further improve upon tests of GR [98]. On the other hand, successful observations of merger-ringdown events with ground-based detectors could allow one to revisit old space-based data and recover sub-threshold events [99], lowering the SNR threshold from 15 to 9 [100] for space-based detectors, which can result in an increased total number of detections [96,99,100]. Finally, the multi-band GW observations of coalescence events have been shown to improve upon the measurement accuracy of several binary parameters, in particular the masses, spins and sky-positions [90,92,96,101].

The red data points in Figure 2 summarize the results determined in [87,88], for the constraint of the eight modified theories of gravity considered here. We observe that, regardless of the PN order at which each effect enters the gravitational waveform, the multi-band observation can improve upon bounds obtained from either the space-based or ground-based detections alone. In particular for the case of dCS gravity, we see that the single-band observations with either detector type fails to satisfy the small coupling approximation (with the exception of DECIGO). Only when utilizing multi-band detections will this approximation become valid, allowing for constraints on $\sqrt{\alpha_{\mathrm{dCS}}}$ to be placed, several orders-of-magnitude stronger than the current constraints. We also observe that several other alternative theories of gravity can be constrained stronger than the current bounds found

in the literature with multi-band observations. We refer to [88] for a comprehensive list of constraints presented here for both single- and multi-band observations.

## 3. Inspiral-Merger-Ringdown Consistency Tests

Let us next test the consistency between the inspiral and merger-ringdown parts of GW signals in a theory-agnostic method. This test, aptly named the inspiral-merger-ringdown (IMR) consistency test [16,17,102–104], allows one to independently compare the two parts of the signal, assuming GR is correct. This is accomplished by estimating the remnant BH's mass $M_f$ and spin $\chi_f$ from each portion of the waveform, and comparing the two. Any statistically significant inconsistencies between the two could be presented as evidence for deviations from GR.

*3.1. Formulation*

In this section, we discuss the formulation and techniques used to carry out the IMR consistency test of GR. We make an assumption that GR is the correct theory of gravity and use the GR waveform templates. To begin, the entire IMR GW signal is divided into the inspiral (I) and merger-ringdown (MR) portions. The transitional frequency between the two is defined to be $f_{\text{trans}} = 132$ Hz for GW150914-like events [17]. Then, one can estimate the four-dimensional probability distributions $P_{\text{I}}(m_1, m_2, \chi_1, \chi_2)$ and $P_{\text{MR}}(m_1, m_2, \chi_1, \chi_2)$ between the BH masses $m_A$ and spins $\chi_A$ from each portion. Such distributions can be obtained with a comprehensive Bayesian analysis as was done in [16,17,102–104], or approximated with the simpler Fisher analysis techniques [52] discussed in Section 2.1, which will be used here. As a result, the probability distributions will take a Gaussian form, centered at the injected masses and spins. Following this, the numerical relativity (NR) fits obtained in [50] in GR allows one to predict the remnant BH's mass $M_f^{\text{I,MR}}(m_1, m_2, \chi_1, \chi_2)$ and spin $\chi_f^{\text{I,MR}}(m_1, m_2, \chi_1, \chi_2)$ from each waveform, entirely from the constituent BH masses and spins. A Jacobian transformation matrix constructed out of such NR fits freely transforms the four-dimensional probability distributions obtained previously into the two-dimensional probability distributions $P_{\text{I}}(M_f, \chi_f)$ and $P_{\text{MR}}(M_f, \chi_f)$. Finally, the consistency between these two distributions provides valuable insight about the gravitational nature of the signal as compared to the assumed theory of GR. Any inconsistencies between the two may point to a modified theory of gravity presenting itself somewhere throughout the entire GW signal.

Typically, the agreement between the two probability distributions above can be measured by once again transforming them together into a joint-probability distribution. We define the new variables $\epsilon$ and $\sigma$ as

$$\epsilon \equiv \frac{\Delta M_f}{\bar{M}_f} \equiv 2\frac{M_f^{\text{I}} - M_f^{\text{MR}}}{M_f^{\text{I}} + M_f^{\text{MR}}}, \qquad \sigma \equiv \frac{\Delta \chi_f}{\bar{\chi}_f} \equiv 2\frac{\chi_f^{\text{I}} - \chi_f^{\text{MR}}}{\chi_f^{\text{I}} + \chi_f^{\text{MR}}}. \tag{9}$$

Here $\Delta M_f \equiv M_f^{\text{I}} - M_f^{\text{MR}}$ and $\Delta \chi_f \equiv \chi_f^{\text{I}} - \chi_f^{\text{MR}}$ describe the differences in the final mass and spin estimates between the inspiral and merger-ringdown signals under the GR assumption, and $\bar{M}_f \equiv \frac{1}{2}(M_f^{\text{I}} + M_f^{\text{MR}})$ and $\bar{\chi}_f \equiv \frac{1}{2}(\chi_f^{\text{I}} + \chi_f^{\text{MR}})$ are the averages between the two. The appendix of [104] describes how the probability distribution of $\epsilon$ and $\sigma$ is derived from the following marginalizations:

$$P(\epsilon, \sigma) = \int_0^1 \int_0^\infty P_{\text{I}}\left(\left[1 + \frac{\epsilon}{2}\right]\bar{M}_f, \left[1 + \frac{\sigma}{2}\right]\bar{\chi}_f\right) \times P_{\text{MR}}\left(\left[1 - \frac{\epsilon}{2}\right]\bar{M}_f, \left[1 - \frac{\sigma}{2}\right]\bar{\chi}_f\right) \bar{M}_f \bar{\chi}_f d\bar{M}_f d\bar{\chi}_f. \tag{10}$$

Finally, the agreement of the resulting probability distribution in the $\epsilon - \sigma$ plane with the GR value of $(\epsilon, \sigma)|_{\text{GR}} = (0, 0)$ determines how consistent the GW signal is with the predictions of GR.

## *3.2. Current Bounds*

Let us now discuss the current status of the IMR consistency tests with the BBH mergers observed thus far. Using a full Bayesian analysis, Abbott et al. [17] performed the IMR consistency test on the LVC catalog of BBH merger events. All such events were found to be statistically consistent with the predictions of GR. While this does not point towards any modifications to GR, such deviations could still potentially be buried within the relatively large statistical noise found within the current generation (O1 and O2) of LIGO-Virgo interferometers. Additionally, Ghosh et al. [102] has discussed testing GR with the IMR consistency test using golden BBH events, as well as adding simulations of modified GR signals in a phenomenological manner [104].

The IMR consistency test has been performed yet again in [87,88] for GW150914-like events, using a simplified Fisher analysis. Figure 4 presents the 90% credible level contours in the $\epsilon - \sigma$ plane. Observe how the two contours for LIGO O1 show good agreement between the Bayesian and Fisher analyses. In order to reveal the resolving power one can gain upon future detections on upgraded interferometers, we focus on the area of these contours. Such resolving power is indicative of how well one can effectively discriminate between GR and non-GR effects entering the gravitational waveform. Once the area of such contours becomes small enough, potential deviations from GR may become highlighted. The top portion of Table 2 compares these areas for the LIGO O1 Bayesian and Fisher results. Observe that the resulting areas agree very well with each other, up to $\sim 10\%$. This indicates that the Fisher analysis IMR consistency tests presented in [87,88] can be trusted to agree somewhat well with their Bayesian counterpart.

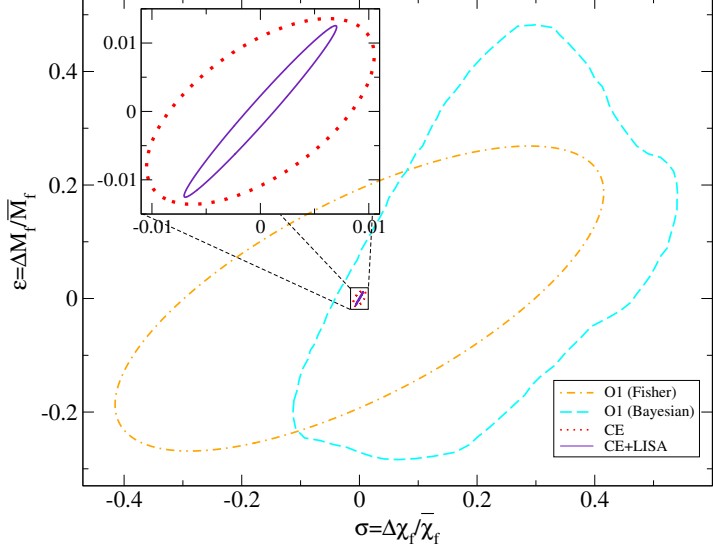

**Figure 4.** The 90% credible region contours of the transformed probability distributions in the $\epsilon - \sigma$ plane, describing the consistency of the remnant mass and spin general relativity (GR) predictions between the inspiral and merger-ringdown waveforms for GW150914-like events. Here we display the results for LIGO O1 (Fisher [87,88] and Bayesian [17] for comparison), CE, and the multi-band observation of CE and LISA. The areas of such confidence regions are displayed in Table 2, and show the following: (i) good agreement within $\sim 10\%$ between the Fisher and Bayesian analyses, (ii) three orders-of-magnitude improvement from LIGO O1 to CE, and (iii) up to an additional order-of-magnitude improvement with multi-band observations. This figure is taken and edited from [88].

## *3.3. Future Bounds*

In this section, we discuss the future prospects for the IMR consistency test, with upgraded third-generation ground-based GW detectors CE. We do not consider space-based interferometers

as they fail to probe the merger-ringdown portion of GW150914-like events, which makes such interferometers incompatible with the IMR consistency test. However, we refer our readers to a work by Hughes and Menou [105], where they described the compatibility of this test with supermassive BBHs observed on space-based detectors.

We summarize our results in Figure 4 and Table 2. Observe that detections of future GW150914-like events by CE can increase the effective non-GR resolving power by up to three orders of magnitude. Such an increase in discriminating power could potentially shed light on any minuscule deviations from GR which could currently be hiding within the detector noise.

*3.4. Multi-Band Bounds*

Here, we discuss how one can further improve upon the IMR consistency test presented in the previous section by making use of multi-band observations between space- and ground-based detectors. While space-based detectors can not observe the merger-ringdown signal for GW150914-like events, they can indeed probe the early inspiral of such events. The multi-band IMR consistency test is performed by first combining the inspiral signal from both the ground-based detector CE, and space-based detectors such as LISA, TianQin, B-DECIGO, and DECIGO. The merger-ringdown portion of the IMR consistency test can then be obtained from the ground-based detector CE alone. The remainder of the test proceeds as before, allowing us to place contours in the $\epsilon - \sigma$ plane for multi-band observations.

Again, the results are summarized in Figure 4 and Table 2 for the multi-band observations between CE and LISA. The former shows the resulting probability distributions in the $\epsilon - \sigma$ plane for such combined multi-band signals, as obtained in [87,88]. Further, the bottom portion of Table 2 presents the resulting areas of the 90% confidence regions from the multi-band observations between the ground-based detector CE and space-based detector LISA. Observe how, in addition to the three-order-of-magnitude improvement made for the future detector CE alone, improvement of an additional factor of about seven can be made by further considering multi-band observations. Moreover, we also found that the multi-band observations with other space-based detectors TianQin, B-DECIGO, and DECIGO, show similar multiplicative improvements in the range of seven to ten [87,88]. Such large improvements may prove to be crucial for future GW observations in highlighting potential deviations from GR which may be small enough to not be visible through CE observations alone.

**Table 2.** Resulting areas of the 90% confidence ellipses from the $\epsilon - \sigma$ posterior distributions for GW150914-like events found in Figure 4, as obtained in [87,88].

| Detector | 90% Area |
|----------|----------|
| LIGO O1 (Fisher) | 0.25 |
| LIGO O1 (Bayesian) [17] | 0.29 |
| CE | $3.6 \times 10^{-4}$ |
| LISA+CE | $5.0 \times 10^{-5}$ |

## 4. Conclusions and Open Questions

In the present communication, we have reviewed the present and future considerations for testing GR with gravitational waves. Non-GR effects may only become actively dominant in extreme-gravity regimes, such as the coalescences of orbiting BHs and/or NSs, which may be effectively probed through the gravitational wave observations of such events. To date, 11 confirmed events have been detected [13–15], and none have thus far been identified to deviate from Einstein's GR [16,17]. However, hope still exists in finding such deviations from GR — these effects, however small they may be, could very well be hidden within the relatively large statistical uncertainties dominant in the current LIGO/Virgo infrastructure. For this reason, many future space-based and ground-based GW interferometers have been proposed, planned, and even funded. With detector noises reaching

up to $\sim 100$ times more sensitive than the current LIGO O2 generation of detectors, in both the low- and high-frequency bands, these detectors stand increasingly large chances of probing these elusive effects in the GW signal. With such detectors, the two fronts of GR tests discussed in this document can be pushed even further than ever done before. We showed that constraints found from several parameterized tests of GR can be improved upon by several orders-of-magnitude with future GW detectors, as well as multi-band observations. Further, we showed that the IMR consistency test can gain many orders-of-magnitude improvement in the resolving power between GR and non-GR effects with such considerations. Together, these improvements can push the bounds formed on many proposed modified theories of gravity.

We end this article by listing several issues that need to be improved further:

1. *Higher PN corrections*: In many cases, the mapping between the ppE parameters and theoretical constants are known only to the leading PN order. However, as compact binaries come close to coalescence, the PN approximation breaks down, and thus it is important to derive and implement higher PN corrections.
2. *Merger-ringdown corrections*: One also needs to include non-GR corrections in the merger-ringdown phases to have complete waveform templates in theories beyond GR. To do so, one needs to to carry out numerical relativity simulations of binary mergers in such theories. Several groups are making progress in this direction [106–114]. Another approach is to extend the effective-one-body waveforms to non-GR theories [115–117].
3. *Precessing/eccentric orbits*: We have focused on spin-aligned binaries with circular orbits. It would be important to extend the analyses described here to more exotic binaries with strong spin-precession [118,119] and largely eccentric orbits.
4. *Cosmological screening*: If one wants to test theories motivated from cosmology, one may need to consider how screening mechanisms affect the GW emission from compact binaries [120–122].
5. *Stacking*: In the future, we expect to have thousands of detections. Thus, we need to study how much improvement one gains in terms of tests of GR with GWs by appropriately stacking multiple events [61,89,123–126].
6. *Sensitivities*: Additional radiation in non-GR theories, such as scalar or dipolar radiation, is typically controlled by the sensitivities of compact bodies [27,127–131]. Currently, BH sensitivities have not been calculated yet in e.g., Einstein-Æther theory and NS sensitivities in this theory need to be revisited within the allowed parameter region after GW170817 [132].

**Author Contributions:** Conceptualization, K.Y.; Calculation, Z.C.; Writing, Z.C. & K.Y.; Visualization, Z.C.; Supervision, K.Y.; Funding Acquisition, K.Y.

**Funding:** This research was funded by NSF Award PHY-1806776, COST Action GWverse CA16104 and JSPS KAKENHI Grants No. JP17H06358.

**Conflicts of Interest:** The authors declare no conflict of interest.

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
