# Peer review of "Parameterized and Consistency Tests of Gravity with GravitationalWaves: Current and Future†"

_proceedings, 2019_

Round 1

Reviewer 1 Report

The paper serves as a valuable review on the current status of testing general relativity in the strong and dynamical regime by gravitational waves, with the emphasis on two approaches, namely the parametrized waveform tests and the inspiral-merger-ringdown consistency tests. It projected the future results that can be obtained from the third generation ground-based detectors, and multi-band observations that involve space-borne detectors. I recommend this paper to be accepted by the MDPI-preceedings after the minor concerns listed below are addressed. 

1) The first paragraph of the introduction would be more readable if references in these open questions (line 20-23) are cited. 

2) (In line 49) For the general concept and preliminary design of TianQin detector, a more appropriate reference may be the one in the link below:

http://adsabs.harvard.edu/abs/2016CQGra..33c5010L

3) Provide the reference(s) for the effective Fisher matrix defined in Eq. 7.

4) Provide the references for the solar system experiments and binary pulsar observations used in Figure 1. 

5) (Line 107) Provide relevant references for ‘many previous works’. 

6) In Table 1, provide full name for the abbreviations in the caption: SEP, PI, etc.

7) (Line 262) This sentence may need more attention.  

8) In various places, such as before Eq. 1, line 84 and 91, the parameterized post-Einsteinian formalism are abbreviated as ppE or PPE, please make the abbreviations consistent in the paper.

Author Response

Dear Editor,

We would like to thank the referee for carefully reading our manuscript
and providing valuable comments which have helped us improve the quality
of our paper. As suggested, we have included citations where applicable
throughout the document: open cosmological questions, TianQin design,
effective Fisher Matrix, Solar System and binary pulsar observations,
and previous parameterized GR testing with phase corrections.
We have additionally described the abbreviations in Tab. I, made
consistent the "ppE" abbreviations throughout the paper, and modified
the confusing sentence in the conclusion.
For convenience, we have appended the comments from the referee to the
end of this document, and we have also highlighted all of our newest
updates in boldface blue.

Sincerely,

Kent Yagi and Zack Carson

---------------------------------------
Response from the Referee
---------------------------------------

The paper serves as a valuable review on the current status of testing
general relativity in the strong and dynamical regime by gravitational
waves, with the emphasis on two approaches, namely the parametrized
waveform tests and the inspiral-merger-ringdown consistency tests. It
projected the future results that can be obtained from the third
generation ground-based detectors, and multi-band observations that
involve space-borne detectors. I recommend this paper to be accepted
by the MDPI-preceedings after the minor concerns listed below are
addressed.

1) The first paragraph of the introduction would be more readable if
references in these open questions (line 20-23) are cited.

2) (In line 49) For the general concept and preliminary design of
TianQin detector, a more appropriate reference may be the one in the
link below:
http://adsabs.harvard.edu/abs/2016CQGra..33c5010L

3) Provide the reference(s) for the effective Fisher matrix defined in
Eq. 7.

4) Provide the references for the solar system experiments and binary
pulsar observations used in Figure 1.

5) (Line 107) Provide relevant references for ‘many previous works’.

6) In Table 1, provide full name for the abbreviations in the caption:
SEP, PI, etc.

7) (Line 262) This sentence may need more attention.

8) In various places, such as before Eq. 1, line 84 and 91, the
parameterized post-Einsteinian formalism are abbreviated as ppE or PPE,
please make the abbreviations consistent in the paper.